# Targeting the Cell Wall Salvage Pathway: Dual-Enzyme Inhibition of AmgK and MurU as a Strategy Against Antibiotic Resistance

**DOI:** 10.3390/ijms26157368

**Published:** 2025-07-30

**Authors:** Hwa Young Kim, Seri Jo, Mi-Sun Kim, Dong Hae Shin

**Affiliations:** College of Pharmacy and Graduates School of Pharmaceutical Sciences, Ewha W. University, Seoul 037601, Republic of Korea; vkwm0221@naver.com (H.Y.K.); seri9388@gmail.com (S.J.)

**Keywords:** cell wall salvage pathway, antibiotic resistance, MurU, Congo red, molecular docking

## Abstract

The rise of multidrug-resistant *Pseudomonas aeruginosa* underscores the need for novel therapeutic targets beyond conventional peptidoglycan biosynthesis. Some bacterial strains bypass MurA inhibition by fosfomycin via a cell wall salvage pathway. This study targeted *P. aeruginosa* AmgK (*Pa*AmgK) and MurU (*Pa*MurU) to identify inhibitors that could complement fosfomycin therapy. A malachite-green-based dual-enzyme assay enabled efficient activity measurements and high-throughput chemical screening. Screening 232 compounds identified Congo red and CTAB as potent *Pa*MurU inhibitors. A targeted mass spectrometric analysis confirmed the selective inhibition of *Pa*MurU relative to that of *Pa*AmgK. Molecular docking simulations indicate that Congo red preferentially interacts with *Pa*MurU through electrostatic contacts, primarily involving the residues Arg28 and Arg202. The binding of Congo red to *Pa*MurU was corroborated further using SUPR-differential scanning fluorimetry (SUPR-DSF), which revealed ligand-induced thermal destabilization. Ongoing X-ray crystallographic studies, in conjunction with site-directed mutagenesis and enzyme kinetic analyses, aim to elucidate the binding mode at an atomic resolution.

## 1. Introduction

*Pseudomonas aeruginosa* is a highly adaptable Gram-negative bacterium that thrives in diverse environments, including the soil, wetlands, and coastal and marine habitats, as well as within plant and animal tissues [1]. Over the past century, *P. aeruginosa* has been recognized as a major pathogen responsible for pneumonia, urinary tract infections, wound infections, bloodstream infections, and cystic-fibrosis-related complications. Pneumonia caused by *P. aeruginosa* presents a significant treatment challenge due to its intrinsic resistance to antibiotics, which can ultimately lead to progressive lung damage and organ failure. The rigid peptidoglycan (PGN) structure of the bacterium plays a crucial role in its defense against environmental stressors, providing it with structural integrity and resistance to cell membrane rupture caused by intracellular osmotic pressure [2,3]. The cell envelope of Gram-negative bacteria consists of a periplasmic space and an outer membrane, including both the cytoplasmic membrane (the inner membrane) and the PGN layer. PGN is surrounded by an inner phospholipid layer and an outer lipopolysaccharide layer, forming a semi-permeable barrier with an asymmetrical outer membrane [4]. Each of these structural components plays a critical role in cell stability and facilitates interactions with the extracellular environment [5].

In *P. aeruginosa*, PGN is an essential component of the cell wall, consisting of polysaccharide strands made up of N-acetylmuramic acid (MurNAc) crosslinked with N-acetylglucosamine (GlcNAc) and peptide bridges [6]. The PGN recycling pathway enables the bacterium to reuse these components, a mechanism known to contribute to antibiotic resistance [2,7,8,9]. Within this pathway, the MurU and AmgK proteins play key roles in PGN biosynthesis. This process involves the synthesis of UDP-MurNAc from MurNAc, catalyzed by MurU and AmgK. Specifically, MurU functions as a transition enzyme, transferring MurNAc-1P to UTP, leading to the production of UDP-MurNAc and diphosphate.

Fosfomycin (C_3_H_7_O_4_P) is an old antimicrobial agent that has been reintroduced to combat infections caused by carbapenem-resistant *Enterobacterales* [10]. However, the emergence of fosfomycin-resistant bacterial strains, such as *Salmonella* spp., poses a growing public health threat. Recent studies suggest that this resistance may be due to a cell wall recycling shortcut, allowing bacteria to bypass de novo biosynthesis of PGN. Fosfomycin specifically targets MurA, an enolpyruvyltransferase and a key enzyme in the PGN biosynthesis pathway [11]. Consequently, alternative drugs targeting PGN synthesis may be required to counteract fosfomycin-resistant bacteria.

In this study, we developed a dual-enzyme assay to evaluate MurU and AmgK from *P. aeruginosa* (*Pa*MurU and *Pa*AmgK), key enzymes involved in the PGN recycling bypass pathway. By integrating a mass spectrometry (MS)-based screening approach, we assayed a chemical library to identify inhibitors of these enzymes. Given *P. aeruginosa*’s extensive antibiotic resistance, which arises through genetic mutations and acquired resistance factors [12], our findings provide valuable insights for the development of novel antibiotics aiming to combat fosfomycin-resistant bacteria.

## 2. Results

### 2.1. The Identification of an Effective Nucleotide for PaMurU

To identify the most effective nucleotide for the optimal *Pa*MurU activity, a dual-enzyme assay was conducted. The results demonstrated that UDP was the most efficient nucleotide, exhibiting the highest phosphorylase activity for *Pa*MurU (Figure 1). In contrast, none of the other tested nucleotides induced or displayed significant phosphorylase activity, confirming UDP as the optimal cofactor for the *Pa*MurU function.

### 2.2. Chemical Screening Using a Dual-Enzyme Assay Method

The dual-enzyme assay demonstrated high reproducibility and throughput. This colorimetric method, which detects inorganic phosphate, showed minimal interference from the assay components, supporting its suitability for screening. Among the compounds tested, Congo red and CTAB exhibited significant inhibitory activity (Figure 2).

### 2.3. MS-Based Identification of Inhibitory Compounds

Since the dual-enzyme assay method was unable to determine which enzyme was inhibited by Congo red or CTAB, an MS approach was employed. This method enabled the identification of the enzymes’ activities by detecting their respective reaction products. Using negative ion electrospray ionization MS (ESI-MS), the deprotonated forms ([M-H]^−^) of each product were identified. In the presence of the compounds, the mass spectrum of the reaction mixture displays peaks corresponding to MurNAc (*m*/*z* 292) and MurNAc-α-1P (*m*/*z* 372), which represent the substrate and the product of *Pa*AmgK, respectively (Figure 3). However, the absence of UDP-MurNAc suggests that the *Pa*MurU activity was suppressed by the compounds, indicating their role as *Pa*MurU inhibitors.

### 2.4. IC_50_ Determination of Congo Red and CTAB Against PaMurU

A dose-dependent inhibition analysis was conducted under saturated substrate conditions (0.25 mM ATP, 0.5 mM MurNAc, and 0.5 mM UTP). The results were plotted as the log inhibitor concentration vs. the percentage reactivity, calculated from the absorbance measurements (Figure 4). The IC_50_ values for Congo red and CTAB, determined from the dose–response inhibition curves, were 14.05 µM and 8.25 µM, respectively.

### 2.5. Molecular Docking Study of Congo Red with PaMurU

As a linear, symmetrical bis-azo dye with naphthalene moieties positioned approximately 20 Å apart, Congo red exhibits a distinct molecular architecture that facilitates its interaction with extended protein interfaces. The top-ranked binding pose from our docking analysis revealed that Congo red engaged in a charge-based interaction via two sulfonate groups. One in the N-terminal part of *Pa*MurU forms charge-based interactions with Asp27, Arg28, and Lys38. The other in the C-terminal part interacts with Arg202 (Figure 5). Additionally, the complex is stabilized further by a hydrogen bond between the amino group of one of the naphthalene sulfonates and Asp155, as well as by π-cation interactions between an electron-rich phenyl ring in Congo red and the amino group of Lys196. This binding mode spans both the catalytic and substrate-binding sites, sterically hindering the enzymatic activity.

### 2.6. Ligand-Induced Thermal Stabilization of MurU Demonstrated Using SUPR-DSF

A thermal stability analysis using the SUPR-DSF platform (Applied Photophysics Inc., Charlotte, NC, USA) [13] revealed significant stabilization of MurU in the presence of both UTP and Congo red. The barycentric mean (Bcm) curves (Figure 6a) illustrate that ligand binding shifted the thermal unfolding profile of MurU to higher temperatures compared to those for the ligand-free control. Figure 6b quantifies this shift, with melting temperatures (Tm) increasing from the baseline value observed for unbound MurU. The statistical analysis summarized in Table 1 confirms these observations. The ΔTm values were positive for both ligands, with Congo red and UTP showing significant increases in Tm1. Specifically, the shift in the melting temperature was statistically significant (*p* < 0.001, *n* = 3), indicating enhanced thermal stability due to ligand binding. This suggests that both UTP and Congo red interact directly with MurU and confer conformational stabilization. The individual Bcm curves (Figure 6c–e) further support this conclusion by showing consistent thermal transitions across replicates for each condition. The stabilization effect of Congo corroborates its potential as a MurU inhibitor and provides biophysical evidence of binding.

## 3. Discussion

### 3.1. The Rationale for Targeting AmgK and MurU

The conventional PGN biosynthesis pathway is targeted by several antibiotics, including fosfomycin, which inhibits MurA, the enzyme responsible for synthesizing UDP-MurNAc. By blocking this precursor’s formation, fosfomycin slows PGN synthesis, leading to bacterial cell lysis [14]. Due to its low toxicity and effectiveness, fosfomycin is used to treat urinary tract infections, gastrointestinal infections, and chronic pulmonary infections caused by multidrug-resistant Pseudomonas strains [15]. However, some *P. aeruginosa* bacteria and its multidrug-resistant strains have developed an alternative cell wall salvage pathway that circumvents this inhibition, reducing the efficacy of fosfomycin. This underscores the need for novel inhibitors targeting enzymes in the cell wall salvage pathway, such as *Pa*AmgK and *Pa*MurU, to complement fosfomycin in resistant bacterial strains.

### 3.2. The Development of a Dual-Enzyme Assay for PaAmgK and PaMurU

To investigate the molecular functions of *Pa*AmgK and *Pa*MurU, their genes were cloned, expressed, and analyzed using a newly developed dual-enzyme assay. This assay, based on a malachite green colorimetric method, enables continuous and efficient measurement of both enzymatic activities. A key advantage is its ability to assess enzyme function even when one substrate is not commercially available. The natural substrate for *Pa*MurU, MurNAc-α-1P, is not commercially obtainable. However, it can be generated by *Pa*AmgK from MurNAc, which is commercially available. By integrating both enzymatic reactions into a single detection system, the assay allows for simultaneous measurement of *Pa*AmgK and *Pa*MurU activities. Although a kinase-based assay was initially considered for confirming AmgK activity, a phosphatase-based approach was ultimately selected due to its ability to detect the activities of both enzymes in a single reaction. When combined with mass spectrometry, this method enables efficient identification of compounds that inhibit both targets.

### 3.3. Screening and Validation of Inhibitors Targeting PaAmgK and PaMurU

Using the dual-enzyme assay, the preferred nucleotide for *Pa*MurU was first identified. The results indicated that *Pa*MurU strictly requires UTP for activity, as no enzymatic function was detected with the other nucleotides (Figure 1). Consequently, UTP was included in all subsequent assays. Approximately 232 chemical compounds (Appendix A) were then screened to identify potential inhibitors of *Pa*AmgK and *Pa*MurU. The assay proved to be highly reproducible and rapid, specifically measuring the release of inorganic phosphate = without interference from other assay components. Among the tested compounds, Congo red and CTAB significantly reduced the absorbance in the dual-enzyme system, indicating inhibitory activity (Figure 2). A mass spectrometric analysis was used to confirm the target enzyme. Both compounds were found to inhibit *Pa*MurU, as no peak corresponding to UDP-MurNAc, the enzymatic product of *Pa*MurU, was detected (Figure 3). To confirm the direct interaction between Congo red and *Pa*MurU, a thermal shift analysis was conducted using the SUPR-DSF platform [13]. The resulting ligand-induced thermal stabilization provided biophysical evidence of binding and supported the correlation between conformational stabilization and functional inhibition.

To clarify how binding conformation influences affinity, we first determined the X-ray crystal structure of ligand-free *Pa*MurU (PDB ID: 8HHD). Co-crystallization trials with selected inhibitors are currently underway, and a detailed analysis of the structural data will be presented in a separate publication.

### 3.4. Docking-Derived Lead Compounds for Antibacterial Agent Design

We performed molecular docking studies of Congo red (4-amino-3-[[4-[4-[(1-amino-4-sulfonatonaphthalen-2-yl)diazenyl]phenyl]phenyl]diazenyl] naphthalene-1-sulfonate), the most potent inhibitor identified in this study. The binding mode of Congo red with *Pa*MurU observed is reminiscent of its previously reported interactions with HIV protease [16] and amyloid fibrils [17], as characterized by docking and spectroscopic studies. In HIV protease, the sulfonate groups of Congo red interact with the arginine residues Arg8 and Arg208 at the dimer interface, stabilizing the dye in an extended conformation similar to that seen in *Pa*MurU. NMR studies of amyloid fibrils have confirmed that the sulfonate groups in Congo red engage in charge interactions with the arginine residues at fibril-binding sites, playing a key role in the dye’s amyloidophilic properties. Considering these findings, the presence of two key arginine residues, Arg28 and Arg202, in *Pa*MurU may be critical to its interaction with Congo red.

Congo red’s ability to bind multiple proteins stems from its conformational rigidity in three-dimensional space and the electrostatic interactions of its naphthalene sulfonate groups. It functions as a molecular ruler, requiring two arginine or other positively charged residues positioned approximately 20 Å apart for effective binding. When these conditions are met, Congo red forms a highly stable, high-affinity interaction with extended protein surfaces in a largely non-specific manner. This unique binding characteristic can serve as a foundation for designing specific molecular binders or be enhanced through derivative modifications. Furthermore, these properties have potential applications beyond biochemistry, particularly in nanoscience. Further mutational and X-ray crystallographic studies will be conducted to validate and refine the current predictions.

Congo red and CTAB are known to exhibit cytotoxicity and broad-spectrum activity; however, both have demonstrated antibacterial effects at low micromolar concentrations in previous studies. Congo red has been reported to interfere with bacterial cell wall components and exhibits minimal toxicity in mammalian cells at concentrations below 10 µM in short-term exposure assays [18,19]. CTAB, a quaternary ammonium surfactant, is known for its antimicrobial efficacy against Gram-negative bacteria, including *P. aeruginosa*, with MIC values in the low micromolar range [20,21]. While these compounds are not ideal therapeutic candidates in their native forms, they serve as valuable tool molecules for identifying ligandable sites and probing enzyme inhibition in pathogen-specific targets. The absence of homologous human enzymes for *Pa*MurU and *Pa*AmgK also supports the feasibility of selective targeting despite compound promiscuity. These findings justify their current use for mechanism-based validation, with future work aiming to improve the specificity and reduce cytotoxic effects through structural optimization and analog development.

## 4. Materials and Methods

### 4.1. Materials and Reagents

The components of the Luria–Bertani (LB) medium were obtained from Condalab (Madrid, Spain). Ampicillin, glycerol, and isopropyl-β-d-1-thiogalactopyranoside (IPTG) were purchased from Sigma (St. Louis, MO, USA). For protein purification, Tris-HCl, NaCl, imidazole, phenylmethylsulfonyl fluoride, and DNase I were obtained from Sigma (St. Louis, MO, USA). The ultrasonic cell disruptor (Digital Sonifier 450) was purchased from Branson (USA). The ÄKTA explorer system and the His-Trap column were from GE Healthcare (Piscataway, NJ, USA). The malachite green reagent components and the other reagents used in the malachite green assay and inhibition studies were purchased from Sigma (St. Louis, MO, USA). The 0.22 μm PVDF syringe filter was obtained from Youngin Frontier (Daejeon, Republic of Korea). The ethanol, formic acid, and acetonitrile used in the MS analysis were purchased from Sigma (St. Louis, MO, USA). The Agilent 1200 HPLC system, the Agilent 6230 LC/MS TOF with an electrospray ionization source, and the calibration solution (G1969-85001) were from Agilent Technologies (Santa Clara, CA, USA). The Agilent MassHunter Data Acquisition software (version B.03.01) and the Agilent MassHunter Qualitative Analysis software (version B.08.00) were used for data processing. The Schrödinger software suite (Maestro, version 14.2) was licensed from Schrödinger, LLC. (New York, NY, USA). The SUPR-DSF platform used for thermal unfolding analysis was obtained from Applied Photophysics Inc. (Charlotte, NC, USA).

### 4.2. Protein Expression and Purification of PaMurU and PaAmgK

The coding sequences of *Pa*MurU (NCBI https://www.ncbi.nlm.nih.gov/. Ref. seq. VVH83045.1) and *Pa*AmgK (NCBI Ref. seq. Q9I5U1.1) were chemically synthesized by Bioneer (Daejeon, Republic of Korea) and cloned into a bacteriophage T7-based expression vector (pBT7-N-His). The recombinant plasmids were transformed into *E. coli* BL21 (DE3) for protein expression. The *E. coli* BL21 (DE3) cells were purchased from Enzynomics (Daejeon, Republic of Korea). The transformed cells were plated onto Luria–Bertani (LB) agar containing 150 μg/mL of ampicillin, and several colonies were selected and cultured in 10 mL of LB broth with the same antibiotic concentration. To prepare stocks for large-scale expression, 0.85 mL of culture was mixed with 0.15 mL of glycerol and stored at 193 K (−80 °C). Frozen stocks were then revived in 5 mL of LB medium, which was subsequently diluted into 1000 mL of fresh LB medium and incubated at 310 K (37 °C) under shaking until an OD_600_ value of 0.6–0.8 was reached. Protein expression was induced using 1 mM of isopropyl-β-d-1-thiogalactopyranoside (IPTG), and the cultures were incubated further under the same conditions for 16 h. The cells were harvested through centrifugation at 7650× *g* (6500 rpm) for 10 min in a refrigerated centrifuge at 277 K (4 °C).

The cell pellet was resuspended in 30 mL of lysis buffer containing 50 mM Tris–HCl (pH 8.0), 100 mM NaCl, 10 mM imidazole, 1 mM phenylmethylsulfonyl fluoride (PMSF), and 10 μg/mL of DNase I. Cell lysis was performed using an ultrasonic cell disruptor (Digital Sonifier 450, Branson, MO, USA), and the lysate was cleared through centrifugation at 24,900× *g* (15,000 rpm) for 30 min at 277 K (4 °C).

Purification of *Pa*MurU and *Pa*AmgK was performed using an ÄKTA explorer system (GE Healthcare, Piscataway, NJ) with His-Trap affinity chromatography followed by cation-exchange chromatography using a 5 mL Hi-Trap Q column (GE Healthcare). For *Pa*MurU, the Hi-Trap Q column was equilibrated with 20 mM Tris (pH 7.5), and the protein fractions were eluted using a linear NaCl gradient up to 1.0 M, with *Pa*MurU eluted at 0.38 M NaCl. Similarly, for *Pa*AmgK, the column was equilibrated using 20 mM Tris (pH 7.0), and the protein was eluted using the same linear NaCl gradient up to 1.0 M, with *Pa*AmgK eluted at 0.36 M NaCl.

### 4.3. Chemical Screening Using the Dual-Enzyme Assay Method

Approximately 232 chemical compounds (Appendix A.) were screened using a dual-enzyme assay based on the malachite green method [22]. This assay relies on the sugar kinase activity of *Pa*AmgK, which catalyzes the conversion of MurNAc into MurNAc-α-1P using ATP. Subsequently, *Pa*MurU transfers UTP to MurNAc-α-1P, generating UDP-MurNAc and pyrophosphate (PPi). The released PPi is hydrolyzed into two phosphate molecules by inorganic pyrophosphatase (IPP), and the resulting phosphate concentration is quantified using the malachite green method. MurNAc, ATP, and UTP were obtained from Sigma-Aldrich. The malachite green reagent used for phosphate detection was prepared by mixing ammonium molybdate ((NH_4_)_6_Mo_7_O_24_), malachite green solution, and Tween 20 at a 1:3:0.1 ratio. The mixture was filtered using a PVDF syringe filter and allowed to stand at room temperature for 1 h before use.

All of the tested chemical compounds (25 μM) were evaluated for their inhibitory potential by comparing their absorbance readings with a control at 620 nm. The actual absorbance was determined as the difference between the absorbance values of the reaction mixtures with and without *Pa*MurU and *Pa*AmgK. The reaction mixture contained 50 mM Tris–HCl (pH 8.0), 10 mM MgCl_2_, 1 unit of IPP, 0.01% Triton X-100, and 0.0125 mg/mL (1.3 μM) of *Pa*MurU and *Pa*AmgK.

To evaluate the reliability of the screening assay, the Z′ factor was calculated and found to be approximately 0.8 (*n* = 10), indicating that the enzyme inhibition assay was conducted under statistically robust and highly reliable conditions.

### 4.4. The IC_50_ Values of Congo Red and Cetyltrimethylammonium Bromide

The dose-dependent inhibitory effect of Congo red and cetyltrimethylammonium bromide (CTAB), purchased from Janssen, was evaluated. A 40 μL reaction mixture containing 50 mM Tris–HCl (pH 8.0), 10 mM MgCl_2_, 1 unit of IPP, 0.01% Triton X-100, 0.0125 mg/mL of *Pa*MurU, and 0.0125 of mg/mL *Pa*AmgK was prepared with varying concentrations of the compounds (0–24 μM). These mixtures were incubated at room temperature for 1 h.

Control reactions were prepared in parallel, containing identical concentrations of the compounds but lacking *Pa*MurU and *Pa*AmgK, and incubated under the same conditions to serve as blanks. The enzymatic reaction was initiated by adding 0.25 mM ATP and 0.5 mM MurNAc, followed by incubation for 30 min. Subsequently, 0.5 mM UTP was added, and the reaction was incubated further for 30 min. After incubation, 160 μL of the previously prepared malachite green reagent was added to each reaction mixture and left to stand for 10 min.

The absorbance was measured at 620 nm using a microplate spectrophotometer (Spectramax 190, Molecular Devices Corporation, Sunnyvale, CA, USA). The percentage reactivity (% Reactivity) was determined based on the difference in the absorbance between the reaction mixtures with and without *Pa*MurU and *Pa*AmgK. The IC_50_ values of the compounds against *Pa*MurU and *Pa*AmgK were calculated using a nonlinear regression analysis in GraphPad Prism 10.1.1 (GraphPad Software, La Jolla, CA, USA).

### 4.5. The Optimized LC-MS TOF Protocol for an MS Analysis of the Reaction Products

All of the incubated mixtures for the MS analysis were diluted 1:1000 with ethanol, filtered using a PVDF syringe filter with a 0.22 μm pore size, and subsequently injected into an Agilent 1200 HPLC system (Agilent, Santa Clara, CA, USA). The mobile phase consisted of 50% (*v*/*v*) water with 0.1% (*v*/*v*) formic acid and 50% (*v*/*v*) acetonitrile with 0.1% (*v*/*v*) formic acid, operating under isocratic conditions at a flow rate of 0.3 mL/min. The HPLC system was connected to an Agilent 6230 LC/MS TOF (Agilent) equipped with an electrospray ionization (ESI) source in negative ion mode. The optimized operating parameters were set as follows: the capillary voltage at 110 V, the nebulizer pressure at 40 psi, the drying gas flow rate at 11 L/min, the gas temperature at 598 K, the skimmer voltage at 65 V, the octapole RF voltage at 750 V, and the fragment voltage (in-source CID fragmentation) at 175 V. The LC-MS accurate mass spectra were recorded over an *m*/*z* range of 100–1100, with accurate mass calibration performed over a range of *m*/*z* 112.9856–1034.9911 using a calibration solution provided by the manufacturer (G1969-85001, Agilent). The data acquisition was carried out using Agilent MassHunter Data Acquisition software (version B.03.01), and the spectral data were processed using Agilent MassHunter Qualitative Analysis software (version B.08.00).

### 4.6. The Ligand and Target Preparation and the Induced-Fit Docking Methodology

All of the molecular docking and scoring procedures were performed using the Schrödinger software suite (Maestro, version 14.2). The test compounds were obtained from the PubChem (https://pubchem.ncbi.nlm.nih.gov/, accessed on 10 May 2025) database in SDF format and compiled into a single dataset. This dataset was then imported into Maestro, where LigPrep was utilized to generate chemically valid 3D molecular structures, ensuring the appropriate protonation and tautomeric states. For target preparation, the atomic coordinates of the *Pa*MurU crystal structure (PDB ID: 8HHD) were retrieved from the Protein Data Bank (PDB. https://www.rcsb.org/, accessed on 10 May 2025). The structure underwent preprocessing using the Protein Preparation Wizard, which included the removal of crystallographic water molecules and other solvent components, the addition of hydrogen atoms, and restrained energy minimization while maintaining the original bound ligand. Additionally, the ionization states of all compounds were predicted at a physiological pH of 7.0 ± 2.0 using an ionizer module to account for relevant protonation states.

The prepared ligands, optimized for low-energy conformations, were subsequently subjected to induced-fit docking (IFD). Following the methodology outlined by Sherman et al. (2006) [23], the IFD workflow was executed through the Maestro graphical interface. Receptor flexibility was accounted for by sampling residues within a 5.0 Å radius of the ligand-binding site, allowing for local structural rearrangements in the presence of the docked ligands. Structural refinement, including side-chain flexibility and backbone minimization, was performed using Prime to enhance the accuracy of the ligand–receptor interactions. Multiple receptor conformations were generated for each ligand to capture induced-fit effects. The test ligands were then re-docked into receptor conformations that exhibited energy values within 30.0 kcal/mol of the lowest-energy structure. The final ligand poses were evaluated using a combined scoring approach, incorporating both the Prime and Glide Score functions, to assess the binding affinity and interaction stability.

This comprehensive workflow, integrating ligand preparation, target optimization, and induced-fit docking, ensured robust predictions of the ligand–protein interactions with high accuracy.

### 4.7. The SUPR-DSF-Based Thermal Unfolding Assay of MurU with and Without Ligands

Thermal unfolding reactions (10 µL) were carried out in a 384-well PCR plate (HSP3886, Bio-Rad), using centrifuged for 1 min at 1000 rpm to remove air bubbles. Each reaction contained 10 mM Tris–HCl (pH 7.5), 0.5 mM MgCl_2_, and 0.05 mg/mL of MurU, with or without ligands. For ligand-binding conditions, 0.1 mM UTP and 10 µM Congo red were added. The samples were subjected to a temperature ramp from 20 °C to 105 °C at 1 °C/min, and fluorescence emissions were recorded across the 310–360 nm wavelength range. Thermal stability measurements were performed using the SUPR-DSF platform according to the manufacturer’s protocol.

## 5. Conclusions

This study demonstrates that targeting *Pa*AmgK and *Pa*MurU represents a promising strategy for overcoming fosfomycin resistance in *P. aeruginosa*. Through the development of a dual-enzyme assay, we successfully identified Congo red and CTAB as effective inhibitors of *Pa*MurU. The inhibitory activity was validated using biochemical assays, IC_50_ determination, and a mass spectrometric analysis, which confirmed the suppression of UDP-MurNAc formation. Furthermore, the ligand binding was independently corroborated by an SUPR-DSF-based thermal shift analysis, showing significant stabilization of *Pa*MurU in the presence of Congo red. Molecular docking further supported a charge-driven binding mode involving key conserved residues (Arg28 and Arg202). Collectively, these findings not only validate our integrated assay platform but also underscore the potential of Congo red derivatives as mechanistic probes and starting points for the development of novel antimicrobial agents targeting the PGN recycling bypass pathway.

## Figures and Tables

**Figure 1 ijms-26-07368-f001:**
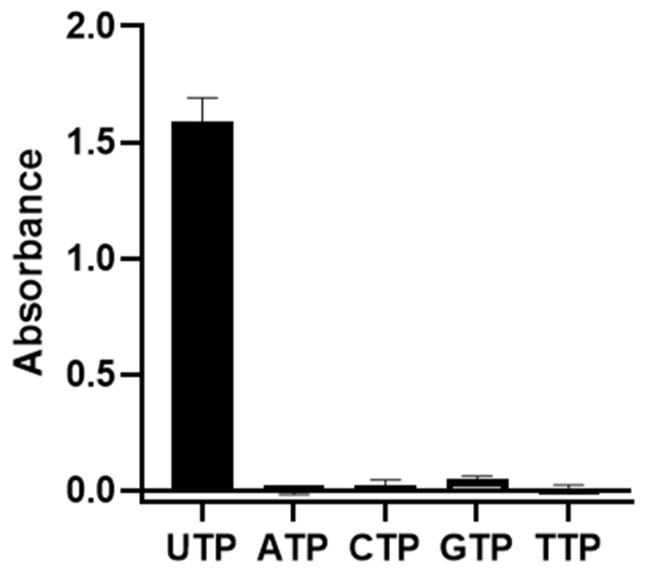
The effect of nucleotides on *Pa*MurU activity. Absorbance measurements indicate UTP as the preferred nucleotide for *Pa*MurU function.

**Figure 2 ijms-26-07368-f002:**
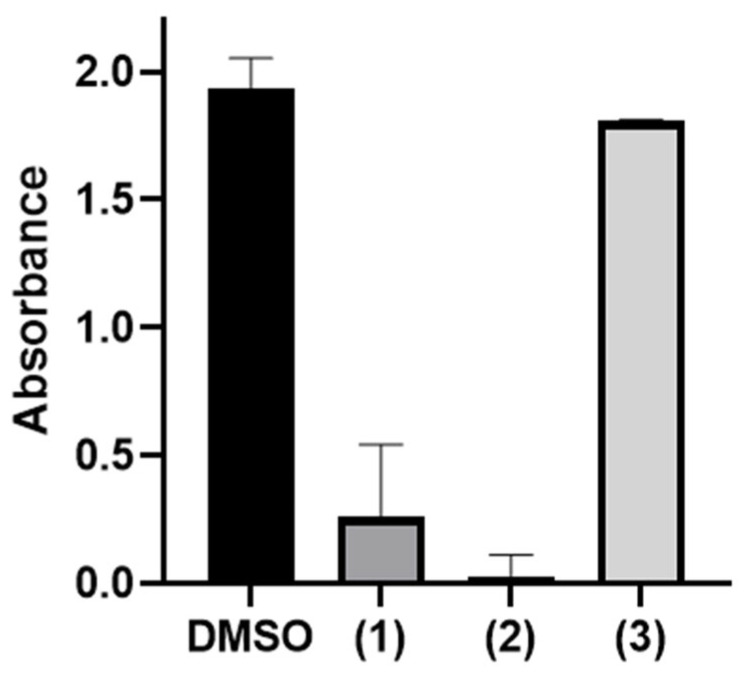
The effect of potent inhibitory compounds on *Pa*MurU activity. Three among the compounds tested, (1) CTAB, (2) Congo red, and (3) D-pantothenic acid, were selected to illustrate their activities. D-pantothenic acid showed no significant effect, with an absorbance similar to that for the DMSO control. Among the tested compounds in this study, Congo red exhibited the strongest inhibitory activity.

**Figure 3 ijms-26-07368-f003:**
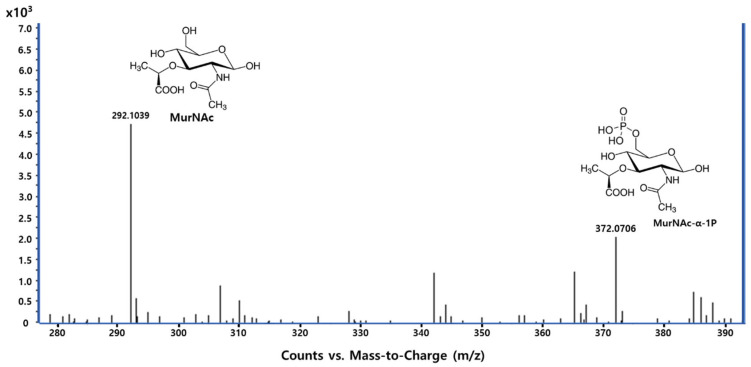
Mass spectrometric identification of the substrate and the product in the presence of Congo red. The Y-axis represents the relative signal intensity of the detected ions.

**Figure 4 ijms-26-07368-f004:**
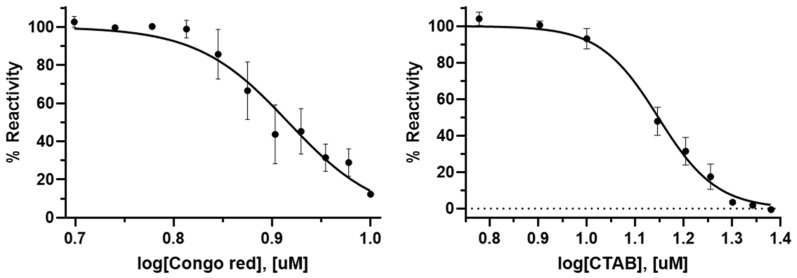
Dose-dependent curves of the inhibition of *Pa*MurU by Congo red and CTAB. The graphs depict the dose-dependent inhibitory effects of Congo red (**left**) and CTAB (**right**) on *Pa*MurU activity. Each data point represents the relative enzymatic activity (% reactivity) compared to that for the control, plotted against the log concentration (µM) of the compounds. The curves show a progressive decline in *Pa*MurU activity with increasing compound concentrations. Data points are expressed as the mean ± standard error of the mean (*n* = 3). The dashed line in the right panel indicates the 0% reactivity baseline.

**Figure 5 ijms-26-07368-f005:**
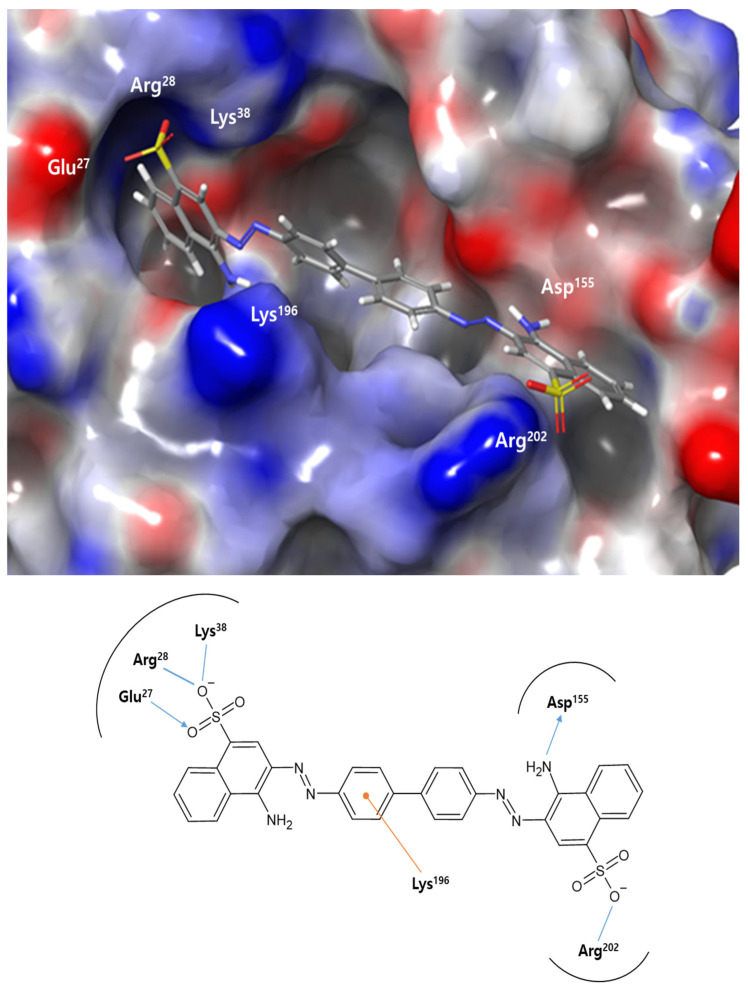
Predicted docking modes of Congo red within the catalytic site of *Pa*MurU. The docking pose illustrates Congo red (gray) on the electrostatic surface potential of *Pa*MurU, where red represents negatively charged regions, blue indicates positively charged regions, and white denotes uncharged areas. A 2D schematic representation of the docked compound with *Pa*MurU is provided. Congo red’s interactions with key side chains in *Pa*MurU are highlighted, with the carbon atom labels referenced in the text. Blue arrows indicate hydrogen bonds, while orange arrows denote π-cation interactions.

**Figure 6 ijms-26-07368-f006:**
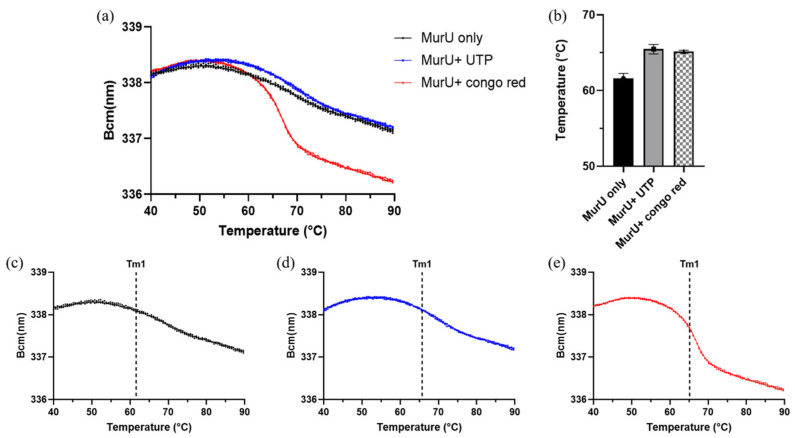
Bcm-based thermal unfolding analysis and Tm comparison of MurU with and without ligand binding. (**a**) Barycentric mean (Bcm) values of MurU were plotted as a function of temperature under three conditions: MurU only (black), MurU + UTP (blue), and MurU + Congo red (red). Each curve represents the mean Bcm value from three replicate measurements. (**b**) Melting temperatures (Tm) were derived from the curves in (**a**) and are summarized as a bar graph (mean ± SD, *n* = 3). (**c**–**e**) Individual Bcm curves from (**a**) are shown separately for each condition, with the corresponding Tm value indicating the thermal transition midpoint.

**Table 1 ijms-26-07368-t001:** The statistical analysis of the Tm1 values for MurU in the presence of UTP or Congo red. Melting temperatures (Tm1) of MurU were measured using SUPR-DSF in the absence or presence of UTP or Congo red. The ΔTm (°C) values were calculated relative to MurU only. Both compounds significantly increased the thermal stability of MurU, as indicated by positive ΔTm values and highly significant adjusted *p*-values (***, *p* < 0.001; *n* = 3).

Sample	Tm1 (Mean ± SD, °C)	Δ Tm (°C)	Mean Diff.	CI (95%)	Abjusted *p*-Value	Sample
MurU only	61.61 ± 0.65	-	-	-	-	-
MurU + UTP	65.47 ± 0.63	+3.86	−3.865	−5.096 to −2.634	0.0002	***
MurU + Congo red	65.16 ± 0.16	+3.55	−3.553	−4.784 to −2.322	0.0003	***

## Data Availability

The authors confirm that the data supporting the findings of this study are available within the article and its Appendix A.

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
