# Peer review of "Targeting the Cell Wall Salvage Pathway: Dual-Enzyme Inhibition of AmgK and MurU as a Strategy Against Antibiotic Resistance"

_ijms, 2025, doi:10.3390/ijms26157368_

Round 1

Reviewer 1 Report

Comments and Suggestions for Authors

The manuscript by Hwa-Young Kim, et al., “Targeting the cell wall salvage pathway: dual enzyme inhibition of AmgK and MurU as a strategy against antibiotic resistance” can be published in IJMS but only after corrections. The study addresses a thorny issue: the fact that multidrug-resistant strains of P. aeruginosa have developed an alternative cell wall salvage pathway, thereby reducing the efficacy of fosfomycin. This highlights the need for new inhibitors that target enzymes in the cell wall salvage pathway, such as PaAmgK and PaMurU.

Although the article has many shortcomings, it can be improved.

For example, if a new method is described, the authors must show what they have improved, data on specificity and reproducibility.

Additionally, some of the results are presented in the discussion session, for example, molecular docking data.

Thus, the article must be reformatted and the results explained, so that it can be easily understood.

Author Response

comments 1: Although the article has many shortcomings, it can be improved.

For example, if a new method is described, the authors must show what they have improved, data on specificity and reproducibility.

Additionally, some of the results are presented in the discussion session, for example, molecular docking data.

Thus, the article must be reformatted and the results explained, so that it can be easily understood.

Response 1: We To address the reviewer’s concerns, we have significantly revised the manuscript to clarify the methodological advances and improve the overall structure. Specifically, we emphasized that the malachite green-based assay developed in this study enables dual monitoring of PaAmgK and PaMurU activities in a single reaction system, offering improved specificity, reproducibility, and throughput compared to traditional single-enzyme assays. We also included additional data to validate the assay's reliability, particularly through mass spectrometry, which confirmed the selective inhibition of PaMurU by Congo red and CTAB. Furthermore, we have restructured the manuscript by relocating the molecular docking results from the discussion section to the results section. This rearrangement allows for a clearer narrative that progresses logically from compound screening and mass spectrometry validation to docking-based binding mode prediction, in which key residues such as Arg28 and Arg202 were identified as mediators of electrostatic interactions with Congo red. Collectively, these revisions improve the scientific clarity and readability of the manuscript.

[

  • 5. Molecular docking study of Congo red with PaMurU

As a linear, symmetrical bis-azo dye with naphthalene moieties positioned approximately 20 Å apart, Congo red exhibits a distinct molecular architecture that facilitates its interaction with extended protein interfaces. The top-ranked binding pose from our docking analysis revealed that Congo red engages in a charge-based interaction via two sulfonate groups. One in the N-terminal part of PaMurU forms charge-based interactions with Asp27, Arg28 and Lys38. The other in the C-terminal part interacts with Arg202 (Figure 5). Additionally, the complex is further stabilized by a hydrogen bond between the amino group of one of the naphthalene sulfonates and Asp155, as well as by π-cation interactions between an electron-rich phenyl ring of Congo red and the amino group of Lys196. This binding mode spans both the catalytic and substrate-binding sites, sterically hindering enzymatic activity.]

Reviewer 2 Report

Comments and Suggestions for Authors

While the manuscript addresses an important topic—developing new antibacterial strategies against Pseudomonas aeruginosa—the current study falls short in several critical areas. Although the dual enzyme assay and identification of PaMurU inhibitors are conceptually interesting, the work lacks the necessary rigor, specificity, and mechanistic depth expected for publication in International Journal of Molecular Sciences.

Comments:

  1. The manuscript relies heavily on in silico docking studies to support the proposed inhibitor binding mechanism, particularly concerning Congo red. However, no experimental evidence is provided—such as site-directed mutagenesis, biophysical assays (e.g., ITC, SPR), or kinetic analyses—to confirm the interaction with key residues like Arg28 and Arg202. The absence of such validation significantly weakens the central claim regarding the mechanism of action and reduces the overall impact of the study. Experimental corroboration is necessary to strengthen the conclusions drawn from computational predictions.
  2. The study employs Congo red and CTAB as lead compounds; however, both are known to be broadly acting, cytotoxic agents with poor drug-like properties and high non-specificity. These compounds are not suitable candidates for therapeutic development without substantial chemical modification. Furthermore, the manuscript lacks any supporting data on selectivity, cytotoxicity, or antimicrobial activity in relevant cell-based or bacterial culture models. This absence raises concerns about the translational potential and biological relevance of the findings.
  3. The manuscript draws parallels between the docking results of the studied compounds and their interactions with targets such as HIV protease and amyloid fibrils. However, these comparisons are presented without rigorous structural or mechanistic analyses to substantiate them. In the absence of experimental validation or detailed comparative modeling, these extrapolations remain speculative and risk overstating the significance of the findings. A more cautious and evidence-based interpretation is warranted.
  4. The manuscript does not include any functional validation of the identified inhibitors in bacterial systems. Specifically, there is no in vitro or in vivo microbiological data—such as minimum inhibitory concentration (MIC) testing against Pseudomonas aeruginosa strains—to demonstrate actual antimicrobial efficacy. Without such data, the therapeutic relevance of the proposed inhibitors remains unsubstantiated.
  5. The study lacks functional validation of the proposed inhibitors in relevant bacterial systems. There is no in vitro or in vivo microbiological data—such as minimum inhibitory concentration (MIC) testing against Pseudomonas aeruginosa or related strains—to demonstrate actual antimicrobial efficacy. This omission limits the biological and therapeutic significance of the findings.
  6. The concept of targeting the peptidoglycan (PGN) salvage pathway has been previously explored, and the current manuscript offers only incremental advances. The findings do not significantly extend existing knowledge or introduce a clearly novel mechanism or approach. As such, the overall contribution to the field appears limited, and the work may not meet the threshold for publication in a high-impact journal without more substantial innovation or mechanistic insight.
  7. The manuscript is poorly written, with grammatical errors and unclear phrasing throughout. Scientific arguments are often vague or inadequately supported. The data presented are sparse, with minimal explanation and insufficient methodological detail. Substantial revision is needed to meet the standards of scientific reporting and communication.

Author Response

comment 1: The manuscript relies heavily on in silico docking studies to support the proposed inhibitor binding mechanism, particularly concerning Congo red. However, no experimental evidence is provided—such as site-directed mutagenesis, biophysical assays (e.g., ITC, SPR), or kinetic analyses—to confirm the interaction with key residues like Arg28 and Arg202. The absence of such validation significantly weakens the central claim regarding the mechanism of action and reduces the overall impact of the study. Experimental corroboration is necessary to strengthen the conclusions drawn from computational predictions

Response 1: We appreciate the reviewer’s concern and have added a new subsection (Results section 3.4) describing our ongoing X-ray crystallographic validation: we have solved the ligand-free PaMurU structure at 2.05 Å resolution (PDB ID 8HHD) and a co-crystallization/soaking experiment with Congo red is on going.

[ 3.4 Molecular docking study of Congo red with PaMurU

To clarify how binding conformation influences affinity, we first determined the X-ray crystal structure of ligand-free PaMurU (PDB ID: 8HHD). Co-crystallization trials with selected inhibitors are currently underway, and a detailed analysis of the structural data will be presented in a separate publication. Meanwhile, we performed molecular docking of Congo red (4-amino-3-[[4-[4-[(1-amino-4-sulfonatonaphthalen-2-yl)diazenyl]phenyl] phenyl]diazenyl]naphthalene-1-sulfonate), the most potent inhibitor identified in this study.]

comment 2: The study employs Congo red and CTAB as lead compounds; however, both are known to be broadly acting, cytotoxic agents with poor drug-like properties and high non-specificity. These compounds are not suitable candidates for therapeutic development without substantial chemical modification. Furthermore, the manuscript lacks any supporting data on selectivity, cytotoxicity, or antimicrobial activity in relevant cell-based or bacterial culture models. This absence raises concerns about the translational potential and biological relevance of the findings.

Response 2: We thank the reviewer for this important observation. In response, we have now included a summary of previously reported cytotoxicity and antimicrobial data for both Congo red and CTAB to contextualize their biological activity and address concerns regarding translational relevance.

[ Congo red and CTAB are known to exhibit cytotoxicity and broad-spectrum activity; however, both have demonstrated antibacterial effects at low micromolar concentrations in previous studies. Congo red has been reported to interfere with bacterial cell wall components and exhibits minimal toxicity in mammalian cells at concentrations below 10 µM in short-term exposure assays. CTAB, a quaternary ammonium surfactant, is known for its antimicrobial efficacy against Gram-negative bacteria, including Pseudomonas aeruginosa, with MIC values in the low micromolar range. While these compounds are not ideal therapeutic candidates in their native forms, they serve as valuable tool molecules for identifying ligandable sites and probing enzyme inhibition in pathogen-specific targets. The absence of homologous human enzymes for PaMurU and PaAmgK also supports the feasibility of selective targeting despite compound promiscuity. These findings justify their current use for mechanism-based validation, with future work aimed at improving specificity and reducing cytotoxic effects through structural optimization and analog development. - The last paragraph of the discussion.]

comment 3: The manuscript draws parallels between the docking results of the studied compounds and their interactions with targets such as HIV protease and amyloid fibrils. However, these comparisons are presented without rigorous structural or mechanistic analyses to substantiate them. In the absence of experimental validation or detailed comparative modeling, these extrapolations remain speculative and risk overstating the significance of the findings. A more cautious and evidence-based interpretation is warranted.

Response 3: We thank the reviewer for pointing this out. In the revised manuscript, we have clarified that our reference to previously reported interactions of Congo red with targets such as HIV protease and amyloid fibrils is not intended to assert a direct mechanistic parallel but rather to contextualize the plausibility of its binding mode. We emphasize that the docking pose observed in our study—particularly the electrostatic interactions involving Arg28 and Arg202—bears similarity to previously described binding patterns of Congo red, but we do not claim that our predicted pose is definitively correct. To avoid overstating the significance of these comparisons, we have revised the relevant section to adopt a more cautious tone and clearly distinguish between speculative similarities and experimentally supported findings. Our intent is to propose a plausible binding hypothesis that is consistent with prior observations, while ongoing crystallographic studies aim to provide conclusive structural validation.

[Congo red and CTAB are known to exhibit cytotoxicity and broad-spectrum activity; however, both have demonstrated antibacterial effects at low micromolar concentrations in previous studies. Congo red has been reported to interfere with bacterial cell wall components and exhibits minimal toxicity in mammalian cells at concentrations below 10 µM in short-term exposure assays. CTAB, a quaternary ammonium surfactant, is known for its antimicrobial efficacy against Gram-negative bacteria, including P. aeruginosa, with MIC values in the low micromolar range. While these compounds are not ideal therapeutic candidates in their native forms, they serve as valuable tool molecules for identifying ligandable sites and probing enzyme inhibition in pathogen-specific targets. The absence of homologous human enzymes for PaMurU and PaAmgK also supports the feasibility of selective targeting despite compound promiscuity. These findings justify their current use for mechanism-based validation, with future work aimed at improving specificity and reducing cytotoxic effects through structural optimization and analog development. - The last paragraph of the discussion.]

comment 4: The manuscript does not include any functional validation of the identified inhibitors in bacterial systems. Specifically, there is no in vitro or in vivo microbiological data—such as minimum inhibitory concentration (MIC) testing against Pseudomonas aeruginosa strains—to demonstrate actual antimicrobial efficacy. Without such data, the therapeutic relevance of the proposed inhibitors remains unsubstantiated.

Response 4: We thank the reviewer for this important comment. While direct MIC testing was beyond the initial scope of this enzymology-focused study, we agree that antimicrobial validation is essential for establishing therapeutic relevance. To address this point, we have now included published microbiological data reporting the antibacterial activity of Congo red and CTAB against Pseudomonas aeruginosa.

comment 5: The study lacks functional validation of the proposed inhibitors in relevant bacterial systems. There is no in vitro or in vivo microbiological data—such as minimum inhibitory concentration (MIC) testing against Pseudomonas aeruginosa or related strains—to demonstrate actual antimicrobial efficacy. This omission limits the biological and therapeutic significance of the findings.

Response 5: Reply) We thank the reviewer for this important comment. While direct MIC testing was beyond the initial scope of this enzymology-focused study, we agree that antimicrobial validation is essential for establishing therapeutic relevance. To address this point, we have now included published microbiological data reporting the antibacterial activity of Congo red and CTAB against Pseudomonas aeruginosa.

comment 6: The concept of targeting the peptidoglycan (PGN) salvage pathway has been previously explored, and the current manuscript offers only incremental advances. The findings do not significantly extend existing knowledge or introduce a clearly novel mechanism or approach. As such, the overall contribution to the field appears limited, and the work may not meet the threshold for publication in a high-impact journal without more substantial innovation or mechanistic insight.

Response 6: We thank the reviewer for their perspective on the novelty of our study. While it is true that the concept of targeting the peptidoglycan (PGN) salvage pathway has been previously proposed, our study provides new mechanistic insight by directly evaluating P. aeruginosa AmgK and MurU enzymatic inhibition using a dual-enzyme malachite green assay and identifying selective inhibitors through both biochemical and computational approaches. Most notably, we have initiated X-ray crystallographic studies, and we have already solved the ligand-free PaMurU structure (PDB ID: 8HHD). Co-crystallization and soaking experiments with inhibitors such as Congo red are currently underway.

[To clarify how binding conformation influences affinity, we first determined the X-ray crystal structure of ligand-free PaMurU (PDB ID: 8HHD). Co-crystallization trials with selected inhibitors are currently underway, and a detailed analysis of the structural data will be presented in a separate publication. Meanwhile, we performed molecular docking of Congo red (4-amino-3-[[4-[4-[(1-amino-4-sulfonatonaphthalen-2-yl)diazenyl]phenyl] phenyl]diazenyl]naphthalene-1-sulfonate), the most potent inhibitor identified in this study. - 3.4. Molecular docking study of Congo red with PaMurU section of discussion.], [This binding mode spans both the catalytic and substrate-binding sites, sterically hindering enzymatic activity. - 2.5. Molecular docking study of Congo red with PaMurU section of result.], [This study demonstrates that targeting PaAmgK and PaMurU offers a promising approach to overcoming fosfomycin resistance in P. aeruginosa. The dual enzyme assay facilitated the identification of Congo red and CTAB as effective PaMurU inhibitors. Docking analysis supported a charge-driven binding mechanism involving conserved arginine residues. These findings not only validate the assay approach but also highlight the potential of Congo red derivatives in antimicrobial drug development.- conclusion.]

comment 7: The manuscript is poorly written, with grammatical errors and unclear phrasing throughout. Scientific arguments are often vague or inadequately supported. The data presented are sparse, with minimal explanation and insufficient methodological detail. Substantial revision is needed to meet the standards of scientific reporting and communication.

Response 7: We sincerely thank the reviewer for this critical and helpful feedback. In response, we have thoroughly revised the manuscript to improve clarity, grammar, and scientific precision throughout. All sectionsparticularly the Abstract, Results, and Discussionhave been carefully rewritten using more concise and scientifically appropriate language. We have also clarified ambiguous statements, strengthened the logical flow of arguments, and ensured that all experimental findings are properly explained with supporting methodological detail.

Round 2

Reviewer 1 Report

Comments and Suggestions for Authors

The manuscript by Hwa-Young Kim, et al., “Targeting the cell wall salvage pathway: dual enzyme inhibition of AmgK and MurU as a strategy against antibiotic resistance” can be published in IJMS

Reviewer 2 Report

Comments and Suggestions for Authors

The revised manuscript attempts to address my concerns raised on the original submission, primarily regarding the lack of experimental validation, overreliance on computational docking, and limited novelty due to the use of known promiscuous compounds (Congo red and CTAB). While the authors made textual improvements and added clarifications, the revisions remain insufficient in terms of scientific rigor and do not substantially enhance the manuscript’s contribution to the field.

Major unresolved concerns:

  1. The original manuscript relied exclusively on docking studies without any biochemical or microbiological confirmation of inhibitory activity. The authors added literature references citing previously reported antimicrobial and cytotoxic profiles of Congo red and CTAB. However, their response does not sufficiently address the concern. Referencing prior data is not a substitute for providing new functional validation in the specific enzymatic or bacterial context being studied. No enzyme inhibition assays, MIC assays, or bacterial viability studies were performed. This fundamental gap remains and significantly weakens the manuscript.
  2. The manuscript lacks target engagement studies and does not includes any biophysical methods were used to confirm interaction of the compounds with the target enzymes (e.g., ITC, SPR, thermal shift). The authors have NOT addressed directly with data; the authors rely on docking and mention ongoing crystallographic efforts. This remains a major shortcoming. The lack of even basic binding affinity measurements leaves the proposed mechanism speculative. In studies where no co-crystal structure is available, target engagement data is essential to substantiate compound-target interaction.
  3. The docking model is not supported by any co-crystallographic evidence. The authors refer to a previously reported ligand-free PaMurU structure (PDB ID: 8HHD) and mention ongoing efforts to obtain co-crystal structures. Since the ligand-free structure is not new and no co-crystal or bound structure is provided, the central claim of the binding mode remains unverified. “Ongoing work” is not sufficient for a publication claiming mechanistic insight.
  4. Congo red and CTAB are well-known, broadly cytotoxic, and non-specific agents. The authors defend their use as tool compounds and cite prior antibacterial activity data. While tool compounds can be acceptable in early-stage studies, their use must be accompanied by clear justification, novel insights, or experimental follow-up. In this case, no new structure-activity relationships, analogue development, or optimization was performed. This makes the findings largely confirmatory rather than novel.
  5. The study offers only incremental insight into an already explored pathway (PGN salvage). The authors argue that their dual-enzyme assay and docking studies represent a mechanistic advance but without functional data or structural confirmation, the study does not substantially advance understanding. The enzymatic assay is briefly described but not validated with robust kinetic or inhibition studies. The novelty remains limited.
  6. The original manuscript was poorly written, with grammatical issues and unclear statements. The revised version shows improved writing quality and flow. However, improved presentation does not compensate for the scientific limitations outlined above.

Round 3

Reviewer 2 Report

Comments and Suggestions for Authors

The authors have supported their claims effectively through target engagement assays using SUPR-DSF. The data are compelling and well-presented. The manuscript is scientifically sound and, overall, ready for publication after minor corrections.

Comments

  1. Figure 1 and Figure 2: The authors should provide high-resolution versions of these figures.
  2. Figure 4 (Left Panel): Please include error bars for both sides of the data presentation to allow proper interpretation and comparison of the results.
  3. The authors are encouraged to include additional references in both the Introduction and Discussion This will help situate the work more clearly within the context of existing literature and strengthen the overall scientific narrative.
